# There's no smoke without fire: Smoking in smoke-free acute mental health wards

**Gabrielle Jenkin**[1]*, **Jacqueline McIntosh**[2], **Janet Hoek**[3], **Krishtika Mala**[3], **Hannah Paap**[1], **Debbie Peterson**[3], **Bruno Marques**[2], **Susanna Every-Palmer**[4]

1 Department of Psychological Medicine, Suicide and Mental Health Research Group, University of Otago Wellington, Wellington, New Zealand, 2 School of Architecture, Victoria University of Wellington, Wellington, New Zealand, 3 Department of Public Health, University of Otago Wellington, Wellington, New Zealand, 4 Department of Psychological Medicine, University of Otago Wellington, Wellington, New Zealand

* gabrielle.jenkin@otago.ac.nz

## Abstract

### Background

People who smoke with serious mental illness carry disproportionate costs from smoking, including poor health and premature death from tobacco-related illnesses. Hospitals in New Zealand are ostensibly smoke-free; however, some mental health wards have resisted implementing this policy.

### Aim

This study explored smoking in acute metal health wards using data emerging from a large sociological study on modern acute psychiatric units.

### Methods

Eighty-five in-depth, semi-structured interviews were conducted with staff and service users from four units. Data were analysed using a social constructionist problem representation approach.

### Results

Although high-level smoke-free policies were mandatory, most participants disregarded these policies and smoking occurred in internal courtyards. Staff reasoned that acute admissions were not the time to quit smoking, citing the sceptres of distress and possibly violence; further, they found smoking challenging to combat. Inconsistent enforcement of smoke-free policies was common and problematic. Many service users also rejected smoke-free policies; they considered smoking facilitated social connections, alleviated boredom, and helped them feel calm in a distressing environment – some started or increased smoking following admission. A minority viewed smoking as a problem; a fire hazard, or pollutant. No one mentioned its health risks.

**Data Availability Statement:** The minimal dataset is provided in the manuscript and the supporting files (the codebook). The data codebook (containing deidentified and aggregated data) is available as a Supporting Information file. In line

with the ethics approval from HDEC ref (17/CEN/94), the full interview transcripts cannot be shared publicly. This is because of their highly sensitive nature and the personal accounts which make them potentially identifiable. For data inquiries please contact the University of Otago Human Ethics Committee, Academic Committees Office, 1st floor, Scott/Shand House, 90 St David's Street, Dunedin, New Zealand, or through the following webpage: https://www.otago.ac.nz/council/committees/committees/HumanEthicsCommittees.html#Contacts/.

**Funding:** This research was funded by a Marsden Fast Start from the Royal Society of New Zealand (contract UOO1623).

**Competing interests:** The authors declare no conflict of interest.

## Conclusion

Psychiatric wards remain overlooked corners where hospital smoke-free policies are inconsistently applied or ignored. Well-meaning staff hold strong but anachronistic views about smoking. To neglect smoking cessation support for people with serious mental illness is discriminatory and perpetuates health and socioeconomic inequities. However, blanket applications of generic policy are unlikely to succeed. Solutions may include myth-busting education for service users and staff, local champions, and strong managerial support and leadership, with additional resourcing during transition phases. Smoke-free policies need consistent application with non-judgemental NRT and, potentially, other treatments. Smoking cessation would be supported by better designed facilities with more options for alleviating boredom, expressing autonomy, facilitating social connections, and reducing distress.

## Introduction

An estimated 13.4% of adult New Zealanders smoke [1, 2]. Although smoking prevalence has declined substantially in recent decades (from 25% in 1996/97), it remains disproportionately high among people with mental illness [2–5], who are twice as likely to be smokers [3, 5–7]. The average mortality gap for people with mental illness is 10–20 years relative to the general population, with a significant proportion of these differing health outcomes likely attributable to smoking [5]. Quitting smoking at age 40 potentially gains six years of life expectancy for those with mental illness [5, 6, 8, 9]; however, quitting success is lower for people with mental illness [6, 7, 10].

To reduce the burden of disease and premature death caused by smoking, in March 2011 the New Zealand (NZ) Government set a goal of Aotearoa (NZ) becoming smoke-free by 2025 [11, 12]. Widely interpreted as meaning smoking prevalence will fall below 5% in all population groups by 2025, the goal has a strong equity focus [12, 13].

Measures supporting the goal include regular excise tax increases (until 2020), the removal of tobacco retail displays, the introduction of standardised packaging, and the expansion of smoke-free areas [12]. This latter provision recognises the World Health Organization's recommendation to ban smoking on healthcare premises and their immediate surroundings [14]. In 2004, NZ became the third country globally to enforce smoke-free indoor workplaces [15]. However, health settings were exempted from requirements to implement smoke-free policies if they met certain conditions [16].

District Health Boards (DHBs) in NZ have shifted towards creating smoke-free environments within mental health wards [3, 17, 18]. However, debate continues over how smoke-free policies work in practice. While the policy has been successfully implemented in indoor hospital settings, smoking continues outside buildings on hospital grounds due to lack of enforcement and, in the case of acute mental health wards, smoking occurs in internal or external courtyards. In some cases, staff are only permitted to smoke offsite while mental health service users have access to designated smoking areas [17, 19, 20].

The enforcement of smoking bans in acute mental health facilities has met with resistance. Deeply embedded historical smoking cultures in mental health settings have made compliance with smoke-free policies challenging [6, 21–24]. Many people who smoke regard smoking as pleasurable and believe it alleviates agitation and anxiety [25]. Historically, some staff have used tobacco to motivate and reward good behaviour; smoking breaks were institutionalised

and used by some staff to establish trust and rapport with patients [22, 26, 27]. The enforcement of smoke-free policies in NZ, as in other countries, has also raised concerns that smoke-free policies in acute mental facilities could discourage patients from presenting for help and created fears that such bans would encroach on personal freedoms [17, 18, 21, 24].

Beliefs that a smoking ban may increase aggression amongst mental health service users in acute mental health care setting also impede smoke-free policy implementation. However, El-Guebaly *et al.*'s 2002 review [28], which examined behavioural changes among service users following enforcement of partial and total smoking bans in mental health facilities, questioned this finding. Of the seven inpatient psychiatric facilities where total bans were enforced, three locked acute units exhibited no increase in aggression, while two acute units reported a decrease in aggression, and only one reported an initial increase in verbal assaults. Of the seven inpatient psychiatric facilities that enforced partial bans, the review only reported the behavioural effects of smoking bans for three inpatient psychiatric facilities, two of which were locked units; all indicated no increase in behavioural disturbances. Some studies of partial bans showed less compliance from heavy smokers, shorter lengths of stay for smokers following the ban, and an initial increase in complaints from service users [28]. In a 2005 review of smoking bans in inpatient psychiatric facilities, Lawn and Pols [29] found 26 relevant studies from US, Canada, and Australia. Overall, regardless of the type of ban enforced in the ten locked units included in the review, few or no adverse events were identified, with no net increase in aggression. Partial bans were mostly enforced in open wards, which again resulted in little to no increase in adverse events, although some conflicts around biased smoking privileges were evident in one setting [29].

Contemporary evidence also does not support arguments that smoking bans in mental health facilities increase violence; when physical or verbal violence does take place, it generally occurs in the initial stages of smoking policy changes at mental health facilities. Any form of violence following smoking bans appears to be temporary and several facilities experienced either decreased violence or no impact following the ban [30, 31]. These findings may be helpful in addressing staff concerns around smoking-bans and violence; nevertheless, policy differences across countries suggest local studies are required to inform both policy and practical support in individual jurisdictions.

Other factors may also contribute to the smoke-free challenge in acute mental health wards. Some research has suggested consistency of smoke-free enforcement by staff is important, with steady enforcement resulting in fewer negative effects than anticipated as service users adapt to smoking bans over time [32–34]. This finding parallels the introduction of smoke-free bars and restaurants, where initial public support for the policy increased from 56% to 69% one year post-implementation [35]. However, successful policy implementation requires staff support; if staff apply the policy inconsistently or themselves continue to smoke in designated smoke-free areas, the policy will not succeed [6, 17, 36]. At worst, inconsistent enforcement will foster covert smoking, concealment of smoking paraphernalia, and smoking in shared outdoor spaces that can put others at risk of second-hand smoke and potentially normalise smoking in those areas [6, 25, 37–40].

Implementing new policies in mental health settings is more complex than general population interventions, given the vulnerable population groups affected. To explore these complexities and how they might be managed, we probed how staff and service users from four adult acute mental health facilities in NZ perceived rules about smoking.

## Methods

This research was part of a larger three-year project 'Design of Acute Mental Health Wards: The New Zealand Experience' examining the architectural design, therapeutic philosophy, and

social regime of adult acute mental health inpatient facilities in NZ. The interview schedule was structured around three main topics: architecture (physical space and sensory aspects); therapeutic environment (recovery, therapy, activities); and social organization (ward rules/ regimes, social relations, and cultural issues).

### Ethics, consultation and locality approvals

The study development included consultation with Otago Ngāi Tahu Research Consultation Committee, whose members review research proposals involving Indigenous Māori in NZ. The Central Health and Disability Ethics Committee reviewed and approved the ethics application (17/CEN/94). We obtained locality consent from the four participating District Health Boards (DHBs). The study protocol is available in the Australian and NZ Clinical Trials Registry (ACTRN12617001469303) http://www.ANZCTR.org.au/ACTRN12617001469303.aspx. As per our research protocol we assured the DHBs that they would be de-identified in any public reporting of results.

### NZ setting

The underpinning philosophy of care for mental health services in NZ, as in many other countries, is the recovery model. Adult acute mental health care in NZ is publicly funded inpatient care provided in 20 acute wards around the country to people with serious or acute mental illness presentations when care can no longer be safely provided in the community. Very few private or non-government acute care facilities exist. Some acute mental health facilities are attached to hospitals and some exist on separate sites. These publicly funded facilities provide short term 'emergency' care for people during a mental health crisis (14–21 day admission target); they are not intended to be long-term residential mental health facilities [41]. Mental health care following discharge from acute inpatient care is provided in the community by non-government organisations (NGOs) and DHB-funded professional mental health support services. These include some limited respite care, supported accommodation, and professional outreach support services provided to people in their homes and communities.

### Case selection

We maximised diversity [42] among the settings included in the study using building age, condition, and location as criteria. Of twenty DHBs, the first four DHBs prioritised for inclusion in our study agreed to participate. These acute mental health wards were spread across NZ; all but one of the acute mental health wards examined was located on a general hospital campus.

### Site visits for interviews

The lead researcher (GJ) visited the four sites multiple times between 2017 and 2018 to recruit participants and conduct interviews.

**Recruitment of staff.**   We sampled staff purposively, to include diverse professions: nurses, health care assistants, social workers, occupational therapists, resident psychiatrists, pharmacists, clinical team leaders, cultural advisors, and consumer advisors.

**Recruitment of service users.**   As per our ethics protocol, we were provided with a list of inpatients from each study site assessed by their lead clinician as competent to consent and well enough to participate, and who were potentially interested in being involved. GJ met with these potential participants and interviewed those who agreed and provided written consent to participate.

**Interview schedule.**   While our semi-structured interview schedule explored multiple aspects of three key topics, this MS reports on responses to the following stem question: 'how do you feel about the rules around smoking on the ward?' We probed responses using prompts, such as 'what do you like/dislike about them?' and 'why do you say that?' While these questions generated rich data on staff members' and service users' perspectives on smoking, many salient comments emerged in response to questions exploring how and where staff and service users spent their time, their use of outdoor spaces, and their perceptions of safety on the ward.

**Interviews.**   All interviews were conducted by GJ, a social scientist and experienced qualitative researcher. Most interviews were conducted face to face on the ward, but ten were conducted by phone. Interviews lasted 30–90 minutes and were transcribed verbatim.

## Analysis

We used a social constructionist problem representation approach to interpret the data [43, 44]. Although designed for analysing policy documents, problem representation is especially useful for examining different perspectives on a policy or social 'problem.' It probes problem representation asking, "What is the problem represented to be?" (WPR). WPR assumes that any discussion of an issue, or problem, is an interpretation that involves judgements, assumptions, and choices. Documenting these interpretations makes the implicit or explicit diagnosis of the 'problem' and its causes transparent and thus exposes implicated policy responses. We present our results using a simplified version [43] of the full WPR approach [44] that considers the problem representation, why it is a problem, and for whom the problem exists, and how the problem might be solved.

All data involving any discussion of smoking was extracted by site from the transcripts by GJ, DP, and HP for coding and analysis; thus, a triple coding process was employed. GJ, HP, SEP, and DP organised the extracted data separately for staff and service users, and developed themes to outline staff members' and service users' perspectives on smoking. We developed key themes iteratively, and refined and re-organised these until all coders agreed on the main themes and sub-themes. Any discrepancies in coding were discussed as a group until an agreement was reached.

## Results

We begin our results by describing the sample and ward setting attributes, then summarise key themes from the interviews. Where themes were common to some staff and service users, we present these in the same section; where themes diverged, we present these separately as staff or service user sub-themes. Evidence supporting the themes is then examined and presented using quotes which, although anonymised, have a code that identifies the case study (Case A, B, C, or D), and whether the participant was a service user or staff member.

### Sample characteristics and ward setting attributes

Across the four wards, 85 interviews were conducted; 43 with service users and 42 with staff (see Table 1 for participant attributes).

Staff participants comprised nurses or nurse managers (n=20), social workers (n=5), psychiatrists (n=4), cultural or consumer advisors (n=4), occupational therapists (n=4), pharmacists (n=2), and a doctor, cleaner, and a music therapist.

Data on diagnoses were not collected and national aggregate data on diagnoses during acute mental health admission are not publicly available; however, a study based in Auckland, NZ found the common discharge diagnoses to be mood disorders, including bipolar disorder (manic, depressive, and mixed episodes) and depression, as well as psychotic disorders, such

**Table 1. Sample characteristics.**

|  | Service users | Staff | Total | Percent |
|---|---|---|---|---|
| Ward A | 10 | 9 | 19 | 22.4 |
| Ward B | 11 | 13 | 24 | 28.2 |
| Ward C | 12 | 11 | 23 | 27.1 |
| Ward D | 10 | 9 | 19 | 22.4 |
| **Total** | **43** | **42** | **85** | **100.0** |
| *Age range* | | | | |
| 20-29 | 13 | 10 | 23 | 27.1 |
| 30-39 | 7 | 6 | 13 | 15.3 |
| 40-49 | 9 | 6 | 15 | 17.6 |
| 50+ | 14 | 20 | 34 | 40.0 |
| **Total** | **43** | **42** | **85** | **100.0** |
| *Gender* | | | | |
| Female | 20 | 27 | 47 | 55.3 |
| Male | 22 | 15 | 37 | 43.5 |
| Transgender | 1 | 0 | 1 | 1.2 |
| **Total** | **43** | **42** | **85** | **100.0** |
| *Ethnicity* | | | | |
| Indigenous Māori | 13 | 8 | 21 | 24.7 |
| NZ European | 27 | 21 | 48 | 56.5 |
| Pacific peoples* | 1 | 3 | 4 | 4.7 |
| Chinese | 0 | 0 | 0 | 0.0 |
| Indian | 0 | 0 | 0 | 0.0 |
| Other | 2 | 10 | 12 | 14.1 |
| **Total** | **43** | **42** | **85** | **100.0** |

*Tongan and Samoan.

as schizophrenia and schizoaffective disorder [45]. Additionally, due to the high demand for acute inpatient care, service users need to present with high acuity to access a psychiatric bed in New Zealand [46]; therefore, the service users participating in this study likely presented with severe mental illness, dominated by mood and psychotic disorders.

## Ward and courtyard characteristics

The characteristics of the four ward case studies, including the courtyards, are given in Table 2. In terms nicotine dependency treatment, acute inpatient wards in NZ normally offer Nicotine Replacement Therapy (NRT) that combines short acting nicotine replacement (lozenges) with long-acting nicotine replacement (21mg nicotine patches). Smokers are provided with cessation advice and support on admission.

All DHBs had similar smoke-free policies prohibiting smoking anywhere on hospital grounds. In practice, however, the approach to smoking varied considerably. Only ward D, the newest building, had successfully implemented a smoke-free policy inside the ward courtyard. In the other units, smoking was allowed, or at least occurred, within the unit courtyard/s.

## Views on smoking and the smoke-free policy

Staff and service users held differing views on smoke-free policies, and we identified three major problem representations. Most staff *and* service users saw the *smoke-free policy* as the

**Table 2. Smoking policy and practice, ward and courtyard characteristics.**

|  | Case A | Case B | Case C | Case D |
|---|---|---|---|---|
| **Smoking policy and practice at data collection** | | | | |
| Policy | Smoke-free inside and outside hospital on wider campus | Smoke-free inside and outside hospital on wider campus | Smoke-free inside and outside hospital on wider campus | Smoke-free inside and outside hospital on wider campus |
| In practise<br>*Outside of ward* | Smoking outside the ward on hospital grounds | Smoking outside the ward on hospital grounds | Smoking outside the ward on hospital grounds | Smoking outside the ward on hospital grounds |
| *Ward courtyards* | Smoking in courtyard | Smoking in courtyard | Smoking in courtyard | Smoke-free courtyards |
| Electronic Nicotine Delivery Systems (ENDS) 'vaping'* | Not allowed | Not allowed | Not allowed | Not allowed |
| **Ward characteristics** | | | | |
| Locked or unlocked** | Locked | Unlocked | Locked | Unlocked |
| Beds | 22 | 64 | 21 | 32 |
| Site situation | On hospital grounds | Not attached to hospital on own park like campus | On hospital grounds | On hospital grounds |
| Number of courtyards*** | 1 | 4 | 1 | 2 |
| **Courtyard characteristics** | | | | |
| Courtyard location | Internal | External facing | Internal | Internal |
| *Surfaces* | Concrete | Grass and garden | Concrete | Concrete, fake grass |
| *Vegetation* | No trees, Basketball hoop, picnic table, potted plants | Green grass, garden, veranda, mesh fencing to outside grounds | Planter box with shrubs | Few shrubs |

*Results section uses the term 'vaping' as this was the term used by participants, however, we refer to Electronic Nicotine Delivery Systems (ENDS) in the discussion.

**Locked and unlocked refers to whether the external door to the outside of the ward was locked or not. By this definition, wards A, C, and D were locked (the lead author could not get out without a staff member unlocking the door), and B claimed to be unlocked, although it was locked on several occasions when the lead author visited.

***Excluding high care and seclusion areas as the high acuity of service users in these spaces prevented access and interviews.

problem; only a minority of both groups (mainly non-smokers and ex-smokers) framed smoking as the problem. The third problem framing related to inconsistent enforcement of the smoke-free rules. Table 3 explains the three problem representations, which we discuss further below.

## I: Smoke-free policy is the problem

Most staff and services users did not support smoke-free wards, with staff explaining that smoking was ubiquitous among mental health service users and difficult to combat, "They smoke, smoke, smoke." (staff A1)

**Difference between policy and practice.** Of the three wards where smoking occurred, staff and service users acknowledged that policy and practice varied:

> *It's meant to be totally smoke-free. The whole hospital is smoke-free, allegedly, but it's not. (staff B9)*

> *You know, the hospital [is] supposed to be a smoke-free environment, I mean it isn't. (service user C6)*

**'Going smoke-free causes violence and aggression'.** Staff from the three wards that had not implemented the smoke-free policy believed that violence and aggression would result from denying service users' requests to smoke and saw this belief as *the* major barrier to going smoke-free:

**Table 3. Summary of staff and service user perspectives on smoking and smoke-free.**

| I: Smoke-free policy is the problem (majority view of both staff and service users) | |
|---|---|
| *Rationale from both staff and service users* | |
| The smoke-free policy does not work in practice | |
| Being smoke-free causes violence and aggression | |
| Smoking has sedative or calming effects (alleviating anxiety and stress) | |
| Smoking is all they/we have (boredom and nothing else to do) | |
| *Staff arguments* | *Service user argument* |
| Acute crisis not the time to quit smoking | Smoking facilitates social connections |
| Smoke-free is difficult to police and takes up staff time | Smoke-free policy infringes on human rights |
| Confrontations are challenging | |
| **II: Smoking is the problem (minority of staff and service users)** | |
| *Staff arguments* | *Service user arguments* |
| Matches and lighters are a safety hazard | Smoking takes over the courtyard |
| Creates exposure to second-hand smoke | Smoking rules are in our best interest |
| **III: Inconsistent rules are the problem (staff and service users)** | |
| *Staff arguments* | *Service user arguments* |
| Differences in smoke-free enforcement causes tensions between staff and services users, and between staff | Differences in enforcement are confusing or unfair |
| | Smoke-free policy can lead to staff locking the unit door even though it is supposed to be unlocked |

> *So, I know that it [smoke-free] was tried a few years ago, before my time, and the assaults went berserk. (staff B9)*

Staff also felt too threatened and intimidated to implement or enforce smoke-free policy:

> *[The smoke-free policy is] something that I would not enforce. I'm not going to get smacked over because someone wants to go and have a cigarette. It's not worth it. You've got to pick your battles. (staff B4)*

Similarly, service users appeared intimidated as they recalled experiences that linked the smoke-free policy with aggression:

> *Well, one guy went off his nut the day before yesterday and started screaming at one of the elderly women because he wanted to go out for a smoke, and they wouldn't open the door until a certain time. So, he lost it, and threatened her. (service user A3)*

Some staff claimed evidence from ward incidents supported their claim that smoking cessation was associated with increased violence:

> *We have a lot of aggression and violence, a lot of that is actually caused by . . . [they] are not allowed to smoke . . . so staff assaults are quite high and there's some stats. (staff C1)*

Managing nicotine withdrawal was therefore viewed as an important part of preventing violence on the ward, which staff did in various ways. Escorting service users outside to smoke was one option for managing the agitation caused by smoking withdrawal, *even on the smoke-free ward*:

*You'd probably just say . . . someone on the floor can go out [to the hospital grounds] with them for five minutes. They can have a cigarette and they can come back in. (staff D5)*

NRT was the most common option for managing agitation due to nicotine withdrawal:

*You can usually de-escalate someone if they're having smoking withdrawal, and we've got the NRT [nicotine replacement therapy], and if we've identified that as the cause of their irritability. (staff D5)*

However, some staff felt NRT had limited effectiveness in reducing the effects of nicotine withdrawal:

*They are offered the NRT . . . But, it's not good enough. It's not satisfying for them [compared to smoking] and—yeah. (staff D8)*

One staff member cited research evidence that vaping reduces violence:

*[We allow vaping] in the courtyard. Both here and HNU [high needs unit]. We are the first DHB in NZ to do it. Australia does it, Canada does it, and UK do it. Research evidence has shown that in psychiatric inpatient units where vaping has been allowed, it's been associated with a 6–8 percent reduction in violence and aggression rates. (staff C7)*

**'Smoking has sedative effects and alleviates stress'.**　　Unsurprisingly, given the significance of the staff and service users' views on the link between nicotine withdrawal and increased violence on the ward, many staff and service users believed that smoking had calming effects, alleviating anxiety and stress:

*For some of our clients it's the only way to cope and keep calm. (staff A9)*

*I would love to give half our patients a cigarette because it's a PRN [pro re nata – medication when necessary] in itself . . . Like a de-escalator in itself . . . instead of antipsychotic . . . or a benzo. (staff D2)*

One staff member told us that the *deliberate* supply of nicotine was a useful calming tool:

*Actually, what they even used to do, which would shock the anti-smoking people, when I was first went there, there was a packet of tobacco and if people were desperate, the nurses could give somebody one, and that really does calm somebody down. (staff C11)*

Service users noted how stress often drove them to smoke in general, and some felt that the stress of being in the acute mental health ward exacerbated their smoking. Being on the ward also contributed to some service users relapsing to smoking as part of socialisation or as self-medication:

*I get a bit tense . . . I bought a cigarette. I'm not addicted but I know these other guys always want to go out for smoke. (service user D8)*

*I gave up smoking, but I've started again . . . I do it when I'm stressed. (service user A7)*

A staff member also commented on smoking uptake among staff as a stress management tool:

*Because there's a lot of staff nurses that smoke . . . stress. Because I had actually stopped smoking for two years, and then I came to work here. (staff A1)*

**'Smoking is all they/we have'.** Both staff and service users talked at times about smoking being 'all they [service users] have.' Staff expressed sympathy that service users had very little and, while it was not clear if they were talking about possessions, people to care for them, or meaningful and pleasurable activities, smoking was conceptualised as one of the few mechanisms of *autonomy* that service users have on the acute ward:

*I think they should be allowed to smoke. I think sometimes that's all they've got left. (staff D7)*

Service users echoed this sentiment:

*They let them smoke outside. You've got to have something in here, you know? (service user A4)*

For most service users, a serious lack of meaningful activity on the acute mental health ward underpinned claims that smoking was all there was. Smoking functioned to alleviate boredom:

*The reason they [other service users] smoke now is because it's boring and that's all there is to do, so that's why they smoke. (service user A6)*

Boredom also drove some service users to smoke more:

*My smoking had increased because of boredom. (service user A9)*

Some cited the lack of activities and boredom as factors that led to smoking relapse:

*[Smoking] That's literally all you've got to do here. And so, I started smoking again when I was brought in here . . . I want something to do so people don't have to smoke. (service user A5)*

**'An acute crisis is not the time to address smoking'.** Some staff thought smoking cessation did not align easily with acute crises and could add to the stress service users experienced:

*Smoking is the hardest thing to give up as far as we know. So why would you make someone who is acutely unwell, give up smoking? (staff D5)*

*They are agitated, they're unwell. A lot of them self-medicate with tobacco, as something to relax them. Yeah. That may not be the best time to make them stop. (staff A6)*

Many staff deemed encouraging smoking cessation as a low priority in the hierarchy of issues that needed to be addressed on the ward:

*You have bigger fish to fry than quitting smoking. You know that's not an issue that we should be trying to tackle, we've got enough issues to deal with. (staff D2)*

*It's good to give them the advice, but really, that's the least of their worries. (staff C11)*

**'Smoke-free is difficult to police and takes up staff time'.**   Staff also talked about the difficulty of 'policing' the smoke-free policy, with service users often smoking covertly in the bathrooms or their bedrooms. Attempts to prevent clandestine smoking, such as removing tobacco and lighters and storing these in the nurses' station, had limited success as service users found novel ways to circumvent these measures:

*The number of lighters! When I'm on nights, and somebody's smoking in their bedroom, which is also against the rules, and I take a lighter off them, and find them the next hour still smoking. I take another lighter, and another lighter, and their cigarettes, and yet they still manage to go to wake somebody else up and get their lighters and cigarettes off them. (staff A3)*

Some staff noted that smoke-free policies caused covert smoking:

*When it was smoke-free, people just smoked in their rooms . . . instead of having someone smoke safely in the courtyard . . . I think there was a curtain lit one day, because he was smoking out the window. (staff C2)*

Some staff also reported how much of their time was spent disregarding the smoke-free policy by escorting service users outside of the ward to the wider hospital grounds to smoke. For service users who were not able to leave the ward unaccompanied, smoking outside was only possible with a willing nurse escort. Many staff had grave concerns about the risk of service users absconding and/or harming themselves if they left the hospital grounds alone:

*. . . [O]bviously they should be going out of the hospital [to smoke], but if an unwell patient [was] walking to the train station, you know what I mean? So, a lot of them smoke in front [of the building], which has, you know, you're not allowed, but staff would almost draw a blind eye because you know, it's either that or that and run away. (staff D1)*

Some staff also lamented high staff turnover, staff shortages, and inadequately trained staff as other factors making smoke-free policy implementation a challenge:

*We're really low on staff and we've such high turnover, a lot of our extra learning opportunities [professional development and training] are getting taken away because we don't have enough people on the ward to do . . . the education. (staff D2)*

**'Smoking confrontations are challenging'.**   Many staff found confiscating tobacco or cigarettes difficult, and potentially likely to result in an unpleasant confrontation, while others suggested enforcing smoke-free policies was a threat to the therapeutic alliance:

*But that is a real challenge . . . we do take the cigarettes off them, but you have to be so careful with how you approach that . . . You might have got on really well with them half an hour ago but right now, they will have a strong reaction to you if you are questioning them about if they've been smoking in their room . . . I support the idea of non-smoking, but . . . (staff D3)*

Another staff member reflected on their previous work experience in an Australian acute mental health ward, where security guards enforced the rules rather than nurses, and suggested this approach protected the therapeutic alliance (anecdotally, we know of at least one ward in NZ that has taken this approach).

Service users who smoked recalled confrontations with other service users who coerced or bullied them to relinquish their tobacco, even on the smoke-free ward:

*He's stolen about, I don't know, 50 bucks [dollars] worth of cigarettes off me, in the space of two weeks. (service user D8)*

**'Smoking facilitates social connections'.**   Service users who smoked also found smoking provided some social benefits; they told us how they chatted and formed social connections when smoking:

*The only time you get together is when you're smoking outside. (service user A9)*

Smoking in the courtyard supported socialising with visiting family members:

*So, they [visiting family] can have a smoke, you see. [Especially] the youngest daughter. (service user B7)*

**'The policy infringes my rights'.**   Some smoking service users framed the issue of smoking as a human rights issue:

*I'm upset 'cause I can't even go outside if say . . . I mean, I still have the rights of people to have a smoke. (service user D2)*

Not being able to smoke in combination with the locked doors contributed to servicer users' sense of imprisonment:

*And being held like we're prisoners . . . I reckon it's against people's rights in a way. . . if no one can actually sit out and have a smoke somewhere. . . I've been only inside in this building; we're not allowed to smoke outside; we're not allowed to go anywhere. (service user D1)*

## II: Smoking is the problem

A small proportion of staff and service users firmly believed that smoking was the problem and thought the DHB was right to adopt a smoke-free policy. Reasons given by staff included safety, the potential for fires, and exposure to second-hand smoke.

**'Matches and lighters are a safety hazard'.**   Staff in all four wards tried to control access to lighters and matches because of concerns about safety due to fires; however, these efforts were not always successful:

*There's been several small fires, usually lit by clients deliberately . . . after we'd had one fire on the open side, in which a patient managed to AWOL [abscond], and the police had to go and collect him from his house, after that, we collected all the lighters in that night, and for the next few nights . . . But the difficulty of getting lighters off patients actually caused several assaults. And then they hadn't just got one lighter. (staff A3)*

Fires also set off the fire alarms, which are very loud, and necessitate evacuation with the risk that some services users may not return to the unit.

**'Second-hand smoking is harmful to others'.**   Staff did not talk specifically about the adverse physical health effects of smoking for those who smoked, but raised concerns about

exposure to second-hand smoke. Staff on two wards, A and B, complained about smoke drifting into their offices or the building generally:

*That's that second-hand smoke thing within five metres of the doors. I'd prefer that we didn't have any smoking in the building, or adjacent to windows and doors. (staff B13)*

**'Smoking monopolises the unit courtyard'.**   Service users' reasons for opposing smoking differed from those identified by staff. Three wards allowed service users to smoke in the courtyards; however, the predominantly non-smoking service users complained that smokers dominated and ruined the courtyards:

*The courtyard was good, but you had to avoid the smokers, and I'm not a smoker. . . it's just a bit, you know, when you had to pick a chair where there weren't smokers around you where you were sitting so you didn't get the passive smoking. (service user A10)*

Service users described the mess caused by butts and ash on the ground, and the circulating smoke, as 'gross,' and suggested various solutions:

*Yeah, the smoko area is . . . [disgusted facial expression] people need to clean up after themselves, there needs to be more strict rules, people coming out and saying, 'Oh clean up, clean up . . . you're not children, you're not three-year olds.' (service user A9)*

Others complained about the mess of covert smoking in other parts of the wards, such as the shared bathrooms:

*The patients don't look after them . . . they smoke in them after the doors have closed . . . They're just real gross. (service user B4)*

Smoke drift's polluting effects extended beyond the courtyards and into other areas of the unit, even in Ward B, which had external facing courtyards. Allowing smoking in the courtyards led non-smoking service users to confine themselves to other areas of the ward:

*I'm a non-smoker, I gave up smoking. I pretty much [spent] a whole lot of time on my own because most people were outside in the courtyard smoking or hanging around in smoking area. I have other health issues, so I don't really want to be hanging around in that, with that smell. It gets in my room. (service user A6)*

Another service user complained that the smoking in the courtyard was a problem for visiting family members:

*There's nowhere for people to bring their family to visit . . . there's two picnic tables and they're usually full of smokers and there's actually nowhere to go with your visitors. (service user A2)*

Although wards have family rooms for visitors to use, these are often small and uninviting, and family members often want to use the outdoor courtyard space for visits.

Solutions offered by service users to the problem of smoking included having designated smoking areas for smokers and smoke-free outdoor areas for non-smoking service users and visiting family:

*You know, if you got rid of the courtyard and made just a little smoking area . . . you could actually have your family there. (service user A2)*

*I'd like smokers to actually be designated somewhere that's inconvenient to them, not inconvenient to people who don't smoke. Because I can't go outside because they smoke. So, I have to be stuck inside. (service user A6)*

Because of differences in the size and layout of the wards, some of these suggestions would require re-design of existing courtyards (Case A), re-purposing of existing courtyards (Case B where there were multiple courtyards), or the addition of another courtyard (Case C).

**Smoking rules are in our best interest.** A minority of service users acknowledged the rules around smoking as in their best interests, not because of smoking's health effects, but because it disturbed sleep and disrupted safety on the ward:

*I'm not addicted but I know these other guys always want to go out for smoke before bed. But the reason you don't is 'cause the nicotine keeps you awake. (service user D8)*

*Because the light and an aerosol can amount to a weapon . . . like a flame torch . . . Which you can remove a service user's face. That's why lighters are separated from service users at night mainly. (service user D3)*

## III: The rules are inconsistent

The final problematisation, shared by many staff and service users regardless of their views on smoking, related to inconsistent rules.

**'Differences in the enforcement of the smoke-free rules cause tension'.** For staff, variable enforcement of the smoke-free rules caused tensions between them and services users:

*So actually, you've got certain staff like me who turn my blind eye to it. Yeah, then you've got other staff who . . . will go out and then say to them you need to put that cigarette out and then it causes . . . [significant problems]. (staff C7)*

Staff also noted that enforcement differences caused tensions between *staff*:

*You'd have a client that is allowed to smoke all day long and then a changeover of shift. The [next] nurse that comes on [says], 'You know this is a no-smoking area, blah, blah, blah'. . . before you know it there's a bit of a tiff between . . . the staff member and the client. And then the staff member feels like that the staff member on the earlier shift had set them up for, for all of this. (staff C10)*

Several staff also reported how making exceptions in some circumstances had caused ongoing problems, as one staff member explained:

*[Going smoke-free] did actually last for about three months, I think. And then of course we had this patient that wasn't allowed any leave at all and was going to be here for quite a long time because they were waiting [for] a rehab bed for him. So, he was getting unofficial permission to smoke, and so everybody else insisted on smoking. So now they all smoke in the courtyard. They have tried to reintroduce the policy, but of course because a lot of our clients are returners, they say, 'Well last time I was in, I was allowed to.' (staff A3)*

However, one staff member conceded that opposition to a smoke-free ward diminished with time:

*When the whole ward was completely smoke-free, and it was enforced, we had issues at the start, but then once people realise you're not getting a cigarette, it becomes less of an issue. But it was tough at the start. We had people attacking us then. But once they realised the only way you're going to get a cigarette is if you work with the team and you can get leave, then they changed their behaviours. (staff A7)*

**'Differences in enforcement are confusing or unfair'.**   For many service users wanting to smoke, inconsistent application of the rules was itself was a source of frustration, resentment, and perceived injustice:

*We are not allowed lighters here [high secure ward] but you're allowed lighters in the other ward. (service user D3)*

Some service users reflected on previous admissions and associated differences in smoke-free policies to changes in locked door policies:

*So, this is the first time that [the door]'s actually been locked and all my other previous admissions I have been able to just walk out the ward and, because there is no smoking policy . . . everybody who was smoking had to go out the front of the hospital. But what was happening was people were doing runners and other people were going off to kill themselves, so. They are now having it that the courtyard, which is a big gigantic ugly concrete structure, is for smoking so that patients don't just leave the ward whenever they want now. (service user A2)*

## Discussion

Our exploration of staff and service users' perspectives on smoking in acute mental health wards revealed multiple complexities that make smoking cessation within these settings highly challenging. We identified a significant discrepancy between policy and practice; in all four wards smoking occurred on the hospital grounds and, in three of the four units, service users smoked inside the courtyard, despite an official smoke-free policy.

Most staff and service users interviewed saw the *smoke-free policy* as the problem. Of the minority who believed that *smoking* was the problem, staff focused on the safety hazards presented by lighters, the exposure to second-hand smoke and, like other research, concern about covert smoking [40]. Surprisingly, none of our participants commented on the undisputed health risks of smoking or the financial burden it imposes. This omission likely reflects their focus on the immediate mental health crisis, which may displace consideration of smoking's long-term impacts [26]. The sad truth is that adults using mental health services in NZ are twice as likely to die before the age of 65 years as other New Zealanders, with most of this premature morbidity attributable to cardiovascular disease, for which smoking is a major risk factor [47, 48]. An admission to an acute ward can be an opportunity for general health screening and intervention in a group who find it difficult to access general healthcare or who are underserved by these services. DHBs have a target of offering at least 90% of people who smoke cessation support [49]; the high rate of smoking among people with mental illnesses means acute mental health care settings provide crucial opportunities for supporting smoking cessation amongst this population.

Like previous studies, we found staff sympathy for those wishing to smoke [27, 39, 40], and concerns about the resourcing and relationship costs of maintaining a smoke-free

environment [17, 26], undermined the implementation of smoke-free polices [26, 27, 40]. Participants held firm but erroneous beliefs that supported 'turning a blind eye' to smoking, including beliefs around violence, mental health, and human rights.

Firstly, staff and service users believed that restricting smoking would result in increased violence. In fact, while anticipatory anxiety about increased aggression arising from the introduction of smoke-free policies has been widespread [26, 29], these concerns are not generally borne out in practice [29]. Studies have demonstrated that the consistent application of smoke-free policies results in fewer negative incidents than feared by staff [26, 32, 37, 39, 40].

Secondly, both service users [40] and staff [17] viewed smoking as having sedative and calming effects. Like some previous research [17, 36], we found that smoking was sometimes used by staff as a behavioural management tool [27]. However, the balance of evidence shows that tobacco is an ineffective strategy for managing stress [50], and has lethal long-term consequences for two thirds of users [51]. In the unit, smoking's assumed calming effects may simply be the alleviation of withdrawal effects.

Smoking does not improve psychiatric outcomes [52, 53]. In fact, there is an association between smoking and suicide; a longer duration of smoking (>40 years versus <10 years) is associated with twice the suicide risk [54]. Recent data are indicative of this being a causal relationship [55]. Treatment of tobacco during psychiatric hospitalisations has also been associated with a decreased likelihood of re-hospitalisation and an increased likelihood of sobriety among smokers with addictive disorders [56–58]. Furthermore, smoking induces cytochrome P450 enzyme activity which increases metabolism of many commonly prescribed psychiatric medications including olanzapine, clozapine, fluoxetine, and fluvoxamine.

Thirdly, consistent with similar research, some service users saw smoke-free policies as infringing their rights and several staff endorsed the view that smoking should be allowed because service users had little else [17]. These beliefs, also found in international research, also warrant scrutiny. While smoking reportedly facilitated social connections between smokers on the ward, it did so at the expense of non-smoking and ex-smoking service users' and staff members' rights. Non-smokers complained they could not use or enjoy the courtyard space, individually or when family members visited, and cigarette smoke and butts degraded the environment. The ubiquity of smoking in the courtyard, and boredom resulting from the lack of alternative activities, resulted in some service users initiating smoking or ex-users relapsing. As previously reported, in an arid inpatient environment, sources of quick gratification – like food or cigarettes – may become substitutes for the hedonic and eudaimonic pleasures that these settings lack [59]. These institutional pressures to smoke have been reported previously [26] and in our view, they reflect a problem with the environment rather than with the smoke-free policy.

Irrespective of their opinions on smoking, both the staff and service users we interviewed complained that inconsistent application of rules corroded relationships. Inconsistencies were detrimental to the success of the smoke-free goal and perceived as unfair [34]. Confusion around smoking prohibitions, for example, where smoking is or is not allowed to occur [39] along with differential treatment of patients, may make it difficult for staff to apply smoke-free policies and lead to conflicts [26] that take time to resolve [29].

## Implications

People with serious mental illness who smoke should be actively supported to quit smoking in the same way as other people who smoke. To neglect offering cessation support to people with serious mental illness is discriminatory and perpetuates the health and socioeconomic burdens that are inequitably borne by those with mental illness. However, the issues are complex and,

as shown in these case studies, a blanket application of a generic smoke-free policy is unlikely to succeed in acute mental health units.

**Targeted myth-busting education.**   Educating staff about the benefits of smoking cessation for people with mental illness should include myth-busting targeted at some of the beliefs shared by our participants.

Staff should be educated about the health, social, and financial costs of smoking for those with serious mental illness, and beliefs on the supposed mental health benefits of smoking should be challenged. Staff should be taught to differentiate between symptoms of nicotine withdrawal and mental illness, understand potential interactions between nicotine and other medications, and provide education and advice to service users who are stopping smoking, including on continued smoking cessation upon discharge. Smoking cessation experts could either train staff or an expert could be embedded in the services to manage the overall smoking cessation programme [17, 27, 40]. While generic policies on smoking cessation exist, this study supports the importance of bespoke packages for mental health services.

**Assertive smoking cessation support.**   Treatments for tobacco dependence are available, well-tolerated, and cost-effective [e.g., 60]. The most successful programs combine pharmacotherapy and behavioural intervention strategies (e.g. physical activity, relaxation, deep breathing) [61]. Tailored treatment programs have been shown to increase attempts to quit and lower levels of nicotine dependence at six months after discharge [62, 63]. Evidence-based treatments available and licensed in NZ include NRT, bupropion, and varenicline.

NRT is used widely in mental health settings, but it would be useful to audit this usage setting against best practice guidelines. Varenicline and bupropion are not routinely prescribed in inpatient wards. After the FDA announced in 2017 that the black box warning for serious neuropsychiatric adverse reactions from the varenicline data sheet in 2017 would be removed, the NZ regulator, Medsafe, issued a prescriber update advising ongoing caution regarding neuropsychiatric adverse reactions [64]. Clinicians still tend to avoid prescribing these drugs during a mental health crisis when people may also have diminished capacity to provide consent. Arguably, given the evidence from the EAGLES trial supporting the safety and efficacy of varenicline and bupropion for those with mental illness, this approach could be reconsidered [65, 66], and these treatments should certainly be offered more assertively to service users in the community.

Some research into smoking cessation support suggests that Cognitive Behavioural Therapy or counselling support in conjunction with pharmacotherapy on the ward may also better support smoking cessation [67].

One recent suggestion has been to allow 'vaping' – use of Electronic Nicotine Delivery Systems (ENDS) in outdoor spaces on the ward. At the time of data collection, the four ward case studies did not allow ENDS use [68]. Research examining ENDS has increased exponentially in recent years but the effectiveness of using e-cigarettes for quitting smoking in the general population remains disputed [69]. While far fewer studies have been conducted with people with serious mental illness, emerging research suggests that ENDS may be a promising intervention to help people from this vulnerable group to switch from smoking [70–72].

A recent open label study examining the effects of ENDs in 40 adult smokers with schizophrenia found that after 12 weeks, 40% had stopped relying on and using traditional cigarettes and the median number of daily cigarettes dropped from 25 to six [72]. After six months, 35% of the participants had stopped smoking traditional cigarettes and continued with vaping and 57.5% already reduced their traditional cigarette use by over half. Participants' quality of life and health also improved, with reductions in blood pressure, weight, and heart rate, and improvements in wellbeing.

In New Zealand, ENDS is not a subsidized pharmaceutical like NRT, thus, the initial cost of purchasing an ENDS device and e-liquids may present a barrier; though, once people switch from smoking, ENDS are typically cheaper to use than smoking.

**Physical design.**   The acute mental health care setting should provide a therapeutic environment that promotes recovery with opportunities for connection, hope and optimism, identity, meaningful activity, and empowerment, along with safety and security [41]. In practice, these units often fall short on alignment of the model of care and ward design with recovery principles. The NZ government has recently committed to a significant funding investment in the rebuild and refurbishment of adult acute mental health facilities in NZ to address these concerns. This investment creates an opportunity to address concerns participants raised about smoking. For example, if courtyards are well-designed, attractive, and smoke-free, they will be therapeutic and encourage use by all, fostering connection and facilitating meaningful activities.

The courtyard of an acute mental health facility is the only outdoor space accessible to all service users and offers therapeutic value via exposure to nature and social interactions. The use of the courtyard as a 'smoking room' has a serious detrimental effect on its usefulness by removing a calming and healthful environment that offers respite from the institutional interiors.

Evidence from the newly-built ward in this study, which has been most successful in the implementation of smoke-free policy, suggests that modern state-of-the-art architectural design combined with a service-user-oriented model of care may have considerable influence on culture and practice.

**Resourcing and support.**   For these strategies to work effectively, staff need the time and resources to provide smoking cessation services. Implementation of smoke-free wards could use a preparation period where staff receive training ahead of smoke-free policy introduction.

NZ acute mental health services have been experiencing year-on-year increases in demand, and services are under-resourced and understaffed, with many of our staff participants describing feeling burnt out [73]. Services may require additional support to persevere through the transition phase of embedding a smoke-free policy.

Maintaining smoking cessation in the community represents a considerable challenge; studies show that relapse rates are high after discharge [27–29]. These findings indicate that additional post discharge smoking cessation support for mental health service users is critical and overdue. Community mental health services and primary care play integral roles and the smoke-free goal should not be abandoned at the hospital gates.

## Strengths, limitations and future research

The core strengths of our study are the rich and detailed data generated from in-depth interviews with 85 participants encompassing staff and service user perspectives. A further strength is the multidisciplinary lens we have brought to this complex issue with our research team comprising social scientists, a social anthropologist, a smoke-free researcher, an academic psychiatrist, a mental health service user academic, and two architects. Collectively we have experience of working, studying, designing, and residing in acute mental health settings. Additionally, our four diverse case studies allowed us to capture factors that may not be found in a single or even two case study design, due to the different architectural layouts of the wards. For instance, second-hand smoke was identified as an issue not just in wards with internal courtyards, but also wards with externally facing courtyards. We believe that many of the results reported here will be generalisable to the wider NZ context, and as many of the nuances in staff and service user perspectives reported here resonate with the wider literature, there is

considerable generalisability to the international context. Another strength was the novel use of the WPR framework used to guide the analysis in this paper.

One limitation is that we cannot triangulate staff members' and service users' comments. For example, the lack of robust regional and national data on violence in acute mental health wards and, as our study showed, the real-life variations in smoke-free policy implementation, make it difficult to contextualise our data. Future work could examine associations between smoking cessation on the ward and violence, and should consider related factors, such as the use of seclusion and locked and unlocked door policies. Another important limitation is that we did not collect staff and service user smoking status or background contextual information on the lead up to smoke-free implementation on acute mental health wards. Future research should include this information. Such work should consider if there has been staff training and education about smoke-free implementation and reasons for a policy change (i.e., critical inequities in smoking outcomes for those with mental illnesses), and how best to provide treat tobacco dependence in this population and setting.

## Conclusion

District Health Boards have become smoke-free in NZ, but acute mental health units remain overlooked corners of the hospital where smoke-free policies are inconsistently applied or ignored. Well-meaning staff in these units continue to hold strong but anachronistic views around smoking that are not upheld by the evidence.

People with serious mental illness who smoke should be actively supported to quit smoking in the same way as others. To neglect this population is discriminatory and perpetuates the health and socioeconomic burdens that are inequitably borne by those with mental illness. Tobacco also harms others. Condoning smoking in an acute setting places all staff and service users at risk due to second-hand smoke exposure. However, the issues are complex and, as shown in these case studies, a blanket application of a generic smoke-free policy is unlikely to succeed. Solutions may include targeted education for service users and staff focussing on myth busting, local champions, leadership, and managerial support with additional resources during the transition phase. Service users found boredom was a major determinant of their desire to smoke in the units – a sad indictment of existing services. People in mental health units require deserve better environments, which may entail ward redesign, providing alternative meaningful activities for service users to alleviate boredom and distress, creating opportunities and spaces for social connection. Nor does supporting people to quit smoking stop and start at the hospital entrance – community mental health services and primary care play important roles in providing assertive smoking cessation support once acute mental health crises have resolved.

## Author Contributions

**Conceptualization:** Gabrielle Jenkin, Debbie Peterson.

**Formal analysis:** Gabrielle Jenkin, Jacqueline McIntosh, Janet Hoek, Hannah Paap, Debbie Peterson, Bruno Marques, Susanna Every-Palmer.

**Funding acquisition:** Gabrielle Jenkin.

**Investigation:** Gabrielle Jenkin, Jacqueline McIntosh, Janet Hoek, Krishtika Mala, Hannah Paap, Debbie Peterson, Bruno Marques, Susanna Every-Palmer.

**Methodology:** Gabrielle Jenkin.

**Project administration:** Gabrielle Jenkin, Susanna Every-Palmer.

**Resources:** Gabrielle Jenkin, Susanna Every-Palmer.

**Supervision:** Gabrielle Jenkin, Jacqueline McIntosh, Susanna Every-Palmer.

**Writing – original draft:** Gabrielle Jenkin, Jacqueline McIntosh, Janet Hoek, Krishtika Mala, Hannah Paap, Bruno Marques, Susanna Every-Palmer.

**Writing – review & editing:** Gabrielle Jenkin, Jacqueline McIntosh, Janet Hoek, Krishtika Mala, Hannah Paap, Debbie Peterson, Bruno Marques, Susanna Every-Palmer.

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
