## [Decision Letter · Decision Letter 0]

25 May 2021

PONE-D-21-12042

There’s no smoke without fire: smoking in smoke-free acute mental health wards

PLOS ONE

Dear Dr. Jenkin,

Thank you for submitting your manuscript to PLOS ONE. After careful consideration, we feel that it has merit but does not fully meet PLOS ONE’s publication criteria as it currently stands. Therefore, we invite you to submit a revised version of the manuscript that addresses the points raised during the review process.

We look forward to receiving your revised manuscript.

Kind regards,

Michael Cummings, PhD

Academic Editor

PLOS ONE

Journal Requirements:

2.When reporting the results of qualitative research, we suggest consulting the COREQ guidelines: http://intqhc.oxfordjournals.org/content/19/6/349. In this case, please consider please provide the interview guide used and please provide additional information regarding the interview guide development process, including the theories or frameworks which were employed.

Furthermore, please reported the date ranges over which participants were recruited form the study and please include in your Methods section (or in Supplementary Information files) the participating hospitals/institutions

Additional Editor Comments :

Please do your best to address reviewer comments.

Reviewers' comments:

Reviewer's Responses to Questions

**Comments to the Author**

1. Is the manuscript technically sound, and do the data support the conclusions?

Reviewer #1: Partly

Reviewer #2: Partly

Reviewer #3: Yes

2. Has the statistical analysis been performed appropriately and rigorously? 

Reviewer #1: N/A

Reviewer #2: Yes

Reviewer #3: N/A

3. Have the authors made all data underlying the findings in their manuscript fully available?

Reviewer #1: No

Reviewer #2: Yes

Reviewer #3: Yes

4. Is the manuscript presented in an intelligible fashion and written in standard English?

Reviewer #1: Yes

Reviewer #2: Yes

Reviewer #3: Yes

5. Review Comments to the Author

Reviewer #1: Study Summary - The authors conducted 85 semi-structured interviews with staff members and service users of four mental health facilities in New Zealand. The interviewers asked about the perceived effectiveness of smoke-free policies in the facilities and the perceived cause of lack of effectiveness. The authors used a social constructionist problem representation approach which "investigates how, and for whom, phenomena are represented as problems, and their implicit or explicit solutions."

Study Results - Results of the interviews indicated that the majority of staff disregarded the facility's smoke-free policies laying out a series of concerns; denying cigarettes leads to patient distress and an increase in aggression and sometimes violence, enforcement problems, and smoking serves as a behavioral management tool. Most patients believed that smoking was important to socializing, reduced boredom, and helped them remain calm. A minority of staff and patients viewed smoking as a problem due to the fire hazard and made common areas less desirable. No one mentioned the health risks associated with smoking. Proposed solutions included increased consistency in enforcement, providing NRT and perhaps vaping, and making the therapeutic milieu more, well, therapeutic.

Strengths - The study focuses on a central disparity issue in tobacco control as individuals with mental illness smoke cigarettes at much higher rates than the general population. They also tend to smoke more cigarettes per day, be more highly addicted, and suffer the chronic disease that inevitably follows. That said, it is unclear what this study actually adds to the discussion given the limitations below.

Weaknesses/Limitations - The authors purposely took an approach in which context is eliminated from the discussion or just ignored. But that is a major flaw of this work and reduces any utility to the fields of tobacco control or mental health treatment. The focus is entirely on the current staff and patients and their perceptions about the law and the source of the "problem." Focusing on staff where the majority are generally non-compliant with the law will give a particular view that the law/policy is the primary problem. A minority of staff see smoking as the problem. I'm not sure what we learn from this that can move the discussion forward. It is exactly the context that was left out of the discussion that might be edifying for those trying to improve compliance with a law that could protect patients who are likely to suffer the negative consequences of smoking and secondhand smoke exposure.

Some of the key contextual elements that we need to know more about include:

How was the policy originally implemented and rolled out? Was there discussion with staff in advance of the new policy? Was there an educational component prior that explained the need for the policy (doubtful since no one mentioned associated health risks as a reason for having the policy)? Was there discussion about staff smoking and offers of free tobacco dependence treatment for staff prior to the law? Was there a reasonable time period from when the policy was announced until the time it became effective? These actions are policy 'best practices" and whether any occurred or not is entirely ignored in this paper. Many of the answers to these questions could explain much of the problems with compliance as well as the perceptions of the staff.

How is tobacco dependence treatment (TDT) for patients managed on these mental health units? Is nicotine dependence treated as less important than the mental health diagnosis or as a secondary substance abuse diagnosis requiring treatment? From the description, treatment of tobacco dependence sounds limited to NRT. What about use of other medications like varenicline, now shown to be safe and effective for individuals with severe mental illness in the EAGLES study. Or the combination of long-acting and short-acting NRT and using higher doses of NRT. Effective use of current TDT does not seem well understood by the staff and it is unclear if the authors understand it either. Tellingly, the authors do consider the use of e-cigarettes though they note the varying concerns on their utility.

Many of the statements from the staff suggest an anachronistic view of tobacco use by patients with mental illness. Suggesting that smoking on a mental health ward can serve as a relaxant, or that it reduces violence (in spite of the authors noting little support for that), or stating that smoking "is the only thing they have," are hopefully falling out of mainstream thinking in treating individuals with severe mental illness. Using smoking as a behavioral management tool (as positive reinforcement for example) is particularly onerous given the long term outcomes of smoking. That staff may join them outside to smoke makes the view more understandable but no more acceptable.

So while the 'data' are interesting, without context in a number of key areas, there is little to guide the improvement of conditions in these wards.

Reviewer #2: Dear Collegues

I very much enjoyed reading your paper - ‘There’s no smoke without fire: smoking in smoke-free acute mental health wards’, of a qualitative study using interviews with 85 patients and staff in 4 wards in NZ and analysed using a social constructionist problem representation approach.

The study extends the literature of smoking among people with mental health problems among in settings in NZ and contributes to the international literature.

A few points you may want to consider

1. The Introduction is comprehensive and provides a useful background to the issue. Re the sentence ‘Quitting smoking at age 40 potentially gains six years of life expectancy for those with mental illness; [5, 6, 8-11]’, whist all these references refer to life expectancy gap between those with and without mental illness, with two of them providing primary data, I’m not sure if all of them either investigated or cited secondary references that quitting at 40 potentially gains six years of life expectancy for those with mental illness. Is it possible to have a check of these refs please?

In addition to El-Guebaly (ref 29) and Lawn’s (refs 26, 30) work, more recent research about staff belief that smokefree polices are related to an increase in violence about violence is not supported by research findings.

Eg Spaducci, G., Stubbs, B., McNeill, A., Stewart, D. and Robson, D. (2018), Violence in mental health settings: A systematic review. Int J Mental Health Nurs, 27: 33-45. https://doi.org/10.1111/inm.12425

Robson et al, Effect of implementation of a smoke-free policy on physical violence in a psychiatric inpatient setting: an interrupted time series analysisLancet Psychiatry 2017; 4: 540–46. http://dx.doi.org/10.1016/S2215-0366(17)30209-2

Methods

Table 2 is very helpful to get an idea of the wards you recruited from. A little more info about what constitutes a ‘site’– for those readers not familiar with how NZ health services are organised, eg were the 4 District Health Boards spread throughout NZ? Were each of the 4 wards part of a mental health hospital or general hospital? Were they acute wards or long stay?

Other than gender, the profile of patients (or staff is not clear), therefore readers won’t be able to judge if the background of the patients you interviewed are similar or different to those they work with; eg did the majority have a severe mental illness eg schizophrenia, bipolar disorder or more a common mental disorder eg – depression , anxiety; were they old or young, had they been in hospital for a long time or a short time? All these variables are likely to influence ones experience of a hospital policy.

Similarly – were the staff young or old? How long had they worked in mental health? You recruited diverse group, though don’t give a breakdown of numbers from each professional role. Had any of the staff received train in smoking cessation?

What was the smoking status of staff and pts?

If none of the info was collected some acknowledgement should be mentioned in the limitations.

Some context about what the policy involves – eg were these hospital policies just related to the environment (which you explain nicely in table 2) or did they also include a tobacco dependence treatment pathway/protocol and if so what did this involve? eg are smokers given NRT on arrival to the ward and thoughout their stay, is it prescribed regularly of do pts have to request it. Is behavioural support provided at all either by ward staff or trained tobacco dependance treatment advisors?

The discussion is comprehensive. Some mention of staff education and training is pertinent when you discuss solutions.

A couple of copy edit points

‘this MS report’s

‘Alternatively, smoking’s assumed calming effects may simply be the alleviation of

withdrawal effects.[put ref above that I deleted here]’

Well done on a great study.

Reviewer #3: This is an interesting article describing tobacco smoking in smoke-free mental health units in New Zealand. The authors apply a social constructionist problem representation approach to interpretation of semi-structured interview data. The framework guiding the analysis is infrequently used in the context of smokefree mental health research, and that is a strength of this paper. The manuscript however, is rather wordy and lacking in some pertinent details, and I feel that this detracts somewhat from its strengths. Some points that the authors might consider:

Abstract: The stated aim of the study is to explore the smoking milieu in the mental health environment. However, the results imply the exploration of other factors in addition to the environment. It is also quite difficult to extract the conclusion from the abstract. It would help if the authors could make clear what they mean by ‘milieu’ – at present it is a little ambiguous.

Introduction: Some of the cited material the authors rely on was published over a decade ago, the inclusion of references to more recent published articles on the implementation and evaluation of smokefree policies would be welcome here. It would also help to orientate international readers if a short summary of the New Zealand mental health landscape was provided. Furthermore, it would be helpful to know if service users are provided with any pharmacological or behavioural support as part of the smokefree policy. A little more detail on the research settings and locality of the findings described would also help the reader to determine contextual similarities and differences to the New Zealand context. This could also be revisited in the discussion. A large number of studies are referenced, but rather than provide a summary of the landscape the introduction would be more compelling if the evidence were used to compile a path forward and a rationale for the present study.

Methods: The authors provide a rather brief overview of the methods they adopted, and leaves the reader with a number of questions. For example, a purposive approach to sampling of the staff group was used, was the same applied to service users? It appears that analysis comprised an iterative thematic approach. What is unclear is the actual process that was undertaken to derive the themes and subthemes. Was analysis undertaken by site/case? How were discrepancies in coding or theme generation resolved? Was a process of double coding employed? The authors should provide a full account of methods. A copy of the interview schedule as supplementary material would also be appreciated.

Results: A table summarising participant characteristic and the inclusion of staff group or service user ward type (locked/unlocked) would help to locate the quotations in terms of perspective. It would also help if the frequencies and/or percentages were reported when discussing majority/minority etc. of participants.

Discussion: The discussion provides details of the New Zealand experience of implementing smokefree policies in acute mental health settings. Whilst the authors experience is in part very similar to that reported by other researchers in other developed countries, there are some local differences and I encourage the authors to expand on the differences; such as the predominance of locked settings and the issue of second-hand smoke in outdoor court yards. It is also helpful that the authors consider the implications of their findings and make recommendations. The limitations might also make mention as to the generalisability of findings within New Zealand and more widely. Perhaps also the authors might consider the impact that those service users who were not or could not be recruited to the study may have on the findings.

The inclusion of e-cigarettes in the discussion is very interesting and relevant. I would encourage the authors to expand on this, particularly in relation to its implementation in the New Zealand context. I would also encourage the authors to identity additional literature from all perspectives in relation to e-cigarettes.

Conclusion: The conclusion seems to lose its way a little. Mention of the more notable findings with potential further avenues for research would I’m sure be appreciated by readers.

6. PLOS authors have the option to publish the peer review history of their article (what does this mean?). If published, this will include your full peer review and any attached files.

Reviewer #1: No

Reviewer #2: No

Reviewer #3: No

---

## [Author Response · Author response to Decision Letter 0]

17 Sep 2021

Dear editors:

The following is the details of the revisions in the manuscript responding to the reviewers' comments. 

Reviewer 1 Comments:

Study Summary - The authors conducted 85 semi-structured interviews with staff members and service users of four mental health facilities in New Zealand. The interviewers asked about the perceived effectiveness of smoke-free policies in the facilities and the perceived cause of lack of effectiveness. The authors used a social constructionist problem representation approach which "investigates how, and for whom, phenomena are represented as problems, and their implicit or explicit solutions."

We have clarified that this research was not intended to be a study on smoke-free implementation on the acute mental health ward. 

Specifically, we have added to the abstract and methods sections, sentences to clarify that the data informing this paper on smoking emerged during the course of a larger programme of sociological research to examine the architectural design, therapeutic philosophy, and social regime of the modern acute mental health unit in New Zealand.

We have noted in the methods now that the interview schedule was structured under three main topics:

1. Architecture (physical space and sensory aspects) 

2. Therapeutic environment (recovery, therapy, activities)

3. Social Organization (ward rules/regimes, social relations, and cultural issues)

The questions from the interview schedule relating to smoking, or smoke-free policy, were not about the perceived effectiveness (or lack of) of smoke-free policies in acute mental health wards which we did not intend to explore. Our data on smoking emerged in response to a number of questions. We have made this clearer in the methods section:

“This MS reports on responses to the following stem question: ‘How do you feel about the rules around smoking on the ward?’ We probed responses using prompts, such as ‘what do you like/dislike about them?’ and, ‘why do you say that?’ While these questions generated rich data on staff members’ and service users’ perspectives on smoking, many salient comments emerged in response to questions exploring how and where staff and service users spent their time, their use of outdoor space, and their perceptions of safety on the ward. 

Study Results - Results of the interviews indicated that the majority of staff disregarded the facility's smoke-free policies laying out a series of concerns; denying cigarettes leads to patient distress and an increase in aggression and sometimes violence, enforcement problems, and smoking serves as a behavioral management tool. Most patients believed that smoking was important to socializing, reduced boredom, and helped them remain calm. A minority of staff and patients viewed smoking as a problem due to the fire hazard and made common areas less desirable. No one mentioned the health risks associated with smoking. Proposed solutions included increased consistency in enforcement, providing NRT and perhaps vaping, and making the therapeutic milieu more, well, therapeutic.

Strengths - The study focuses on a central disparity issue in tobacco control as individuals with mental illness smoke cigarettes at much higher rates than the general population. They also tend to smoke more cigarettes per day, be more highly addicted, and suffer the chronic disease that inevitably follows. That said, it is unclear what this study actually adds to the discussion given the limitations below.

Weaknesses/Limitations - The authors purposely took an approach in which context is eliminated from the discussion or just ignored. But that is a major flaw of this work and reduces any utility to the fields of tobacco control or mental health treatment. 

As noted above, we did not purposely exclude context, rather, the overall study was never designed to focus on smoke-free implementation; however, we agree with the reviewer that such contextual information would be useful in future studies and have noted this in the Discussion under limitations and future research.

“Another important limitation is that we did not collect background contextual information on the lead up to smoke-free implementation on acute mental health wards. Future research could examine, for instance, if there was staff training and education around smoke-free implementation and reasons for a policy change (i.e. critical inequities in smoking outcomes for those with mental illnesses), and how best to provide treat tobacco dependence in this population and setting and NRT support.” 

The focus is entirely on the current staff and patients and their perceptions about the law and the source of the "problem." Focusing on staff where the majority are generally non-compliant with the law will give a particular view that the law/policy is the primary problem. A minority of staff see smoking as the problem. I'm not sure what we learn from this that can move the discussion forward. It is exactly the context that was left out of the discussion that might be edifying for those trying to improve compliance with a law that could protect patients who are likely to suffer the negative consequences of smoking and second-hand smoke exposure.

We are not singling out staff that disagree with the smoke-free agenda, we are reporting the fact that the majority of staff see that smoke-free policy as the problem. What we learn from this paper are the reasons why the majority of staff do not support a smoke-free agenda, even those on the ward where the courtyards were smoke-free. The point about context has been addressed above. We have revisited the discussion, re-writing substantial sections, adding more references and NZ context, and provided recommendations, e.g. education on myth-busting for staff, to move the discussion forward.

Some of the key contextual elements that we need to know more about include:

How was the policy originally implemented and rolled out? Was there discussion with staff in advance of the new policy? Was there an educational component prior that explained the need for the policy (doubtful since no one mentioned associated health risks as a reason for having the policy)? Was there discussion about staff smoking and offers of free tobacco dependence treatment for staff prior to the law? Was there a reasonable time period from when the policy was announced until the time it became effective? These actions are policy 'best practices" and whether any occurred or not is entirely ignored in this paper. Many of the answers to these questions could explain much of the problems with compliance as well as the perceptions of the staff.

We agree that it would be good to have more information on contextual elements, and have added what we know from the NZ context to the paper, in the methods and in the discussion. However, this study was not designed to examine this, and we are unable to obtain accurate information on the rollout of smoke-free policies in each of the four wards due to high staff turnover in these settings. Also, there is a difference, as this paper has shown, between policy and what happens in reality. We have added to the discussion a section highlighting how these factors would be important to collect in future studies. We have added substantially to the discussion, text to help those working in the sector better implement smoke-free policies in the future. 

How is tobacco dependence treatment (TDT) for patients managed on these mental health units? Is nicotine dependence treated as less important than the mental health diagnosis or as a secondary substance abuse diagnosis requiring treatment? From the description, treatment of tobacco dependence sounds limited to NRT. What about use of other medications like varenicline, now shown to be safe and effective for individuals with severe mental illness in the EAGLES study. Or the combination of long-acting and short-acting NRT and using higher doses of NRT. Effective use of current TDT does not seem well understood by the staff and it is unclear if the authors understand it either. Tellingly, the authors do consider the use of e-cigarettes though they note the varying concerns on their utility.

In response to the first question, is nicotine dependence treated as less important than mental health diagnoses, the answer is, yes, by nursing staff in practice. This study showed that staff do not think nicotine dependence is very important to address compared to helping mental health service users survive their mental health crisis. Of course, it should be important, and we have added to the discussion the need to address smoking and all the health inequalities that follow from smoking by those with mental illnesses. 

In response to the query about the use of other medications like Varenicline, we have added some text to the discussion around this. We note that Varenicline and Bupropion are not routinely prescribed in the NZ acute setting, but may be offered subsequently once the person’s mental state has stabilised. We note that after the FDA announced that they would remove the black box warning for serious neuropsychiatric adverse reactions from the Varenicline (Champix) data sheet in 2017, New Zealand Medsafe put out a prescriber update advising ongoing caution (https://www.medsafe.govt.nz/profs/PUArticles/December2017/Varenicline.htm). Prescribers are still inclined to take a conservative approach and avoid prescribing these drugs when someone is experiencing an acute mental health crisis necessitating admission when they may be unable to provide informed consent. 

We have also added prior to the Table that describes ward characteristics, the following sentence about treatment of nicotine dependence on acute mental health wards:

“In terms of treatment for nicotine dependency, the usual approach on acute inpatient wards in New Zealand is to offer Nicotine Replacement Therapy combining short acting nicotine replacement (lozenges) with long acting nicotine replacement (21mg nicotine patches). Smokers are provided with advice and support around quitting on admission.”

Many of the statements from the staff suggest an anachronistic view of tobacco use by patients with mental illness. Suggesting that smoking on a mental health ward can serve as a relaxant, or that it reduces violence (in spite of the authors noting little support for that), or stating that smoking "is the only thing they have," are hopefully falling out of mainstream thinking in treating individuals with severe mental illness. Using smoking as a behavioral management tool (as positive reinforcement for example) is particularly onerous given the long-term outcomes of smoking. That staff may join them outside to smoke makes the view more understandable but no more acceptable.

It is true that many of the statements from staff suggest that they believe smoking provides a number of supposed benefits including reducing stress. This view may be incorrect, and is antiquated; however, it is highly prevalent, nevertheless. We note that staff found it easy to give service users tobacco as a reward for good behaviour, or to alleviate stress, and this has been found in previous research, and staff did not report this as being onerous. We have added section to the discussion around the need to conduct education with staff around smoking cessation on the ward and that this should include myth busting. 

So while the 'data' are interesting, without context in a number of key areas, there is little to guide the improvement of conditions in these wards. 

We have added as much context as we can to the paper, in the methods, results and the discussion. Our discussion in now framed around how to improve the situation and better support smoking cessation amongst mental health services, and smoke free wards.

Reviewer 2 Comments:

Dear Colleagues

I very much enjoyed reading your paper - ‘There’s no smoke without fire: smoking in smoke-free acute mental health wards’, of a qualitative study using interviews with 85 patients and staff in 4 wards in NZ and analysed using a social constructionist problem representation approach.

The study extends the literature of smoking among people with mental health problems among in settings in NZ and contributes to the international literature.

A few points you may want to consider

1. The Introduction is comprehensive and provides a useful background to the issue. Re the sentence ‘Quitting smoking at age 40 potentially gains six years of life expectancy for those with mental illness; [5, 6, 8-11]’, whist all these references refer to life expectancy gap between those with and without mental illness, with two of them providing primary data, I’m not sure if all of them either investigated or cited secondary references that quitting at 40 potentially gains six years of life expectancy for those with mental illness. Is it possible to have a check of these refs please?

In addition to El-Guebaly (ref 29) and Lawn’s (refs 26, 30) work, more recent research about staff belief that smokefree polices are related to an increase in violence about violence is not supported by research findings.

Eg Spaducci, G., Stubbs, B., McNeill, A., Stewart, D. and Robson, D. (2018), Violence in mental health settings: A systematic review. Int J Mental Health Nurs, 27: 33-45. https://doi.org/10.1111/inm.12425

Robson et al, Effect of implementation of a smoke-free policy on physical violence in a psychiatric inpatient setting: an interrupted time series analysis Lancet Psychiatry 2017; 4: 540–46. http://dx.doi.org/10.1016/S2215-0366(17)30209-2

We have checked whether the references cited in the sentence ‘Quitting smoking at age 40 potentially gains six years of life expectancy for those with mental illness; [5, 6, 8-11]’. Two of the resources used primary data (Tam, Warner & Meza, 2016; Lawrence, Hancock & Kisely, 2013), and two are reports (Harker & Cheeseman, 2016; Royal College of Physicians & Royal College of Psychiatrists, 2013). We have removed the two sources that were reviews. As such, the sentence now reads, Quitting smoking at age 40 potentially gains six years of life expectancy for those with mental illness; however, quitting success is lower for people with mental illness. The more recent research suggested by reviewer 2, which conclude that smoke-free policies do not result in an increase in violence, have been added to the introduction. 

Methods

Table 2 is very helpful to get an idea of the wards you recruited from. A little more info about what constitutes a ‘site’– for those readers not familiar with how NZ health services are organised, eg were the 4 District Health Boards spread throughout NZ? Were each of the 4 wards part of a mental health hospital or general hospital? Were they acute wards or long stay?

We have added a subsection ‘NZ Setting’ under Methods, to describe how acute mental health care is organised and provided in New Zealand and reference our earlier paper from this research, which describes the NZ setting in detail. As part of our agreement with the DHB case studies, the DHBs needed to be de-identified. However, we have added to the methods that the 4 District Health Boards were spread throughout New Zealand, and that all but one of the wards were located in a general hospital. 

Other than gender, the profile of patients (or staff is not clear), therefore readers won’t be able to judge if the background of the patients you interviewed are similar or different to those they work with; eg did the majority have a severe mental illness eg schizophrenia, bipolar disorder or more a common mental disorder eg – depression , anxiety; were they old or young, had they been in hospital for a long time or a short time? All these variables are likely to influence ones experience of a hospital policy.

We did not collect data on diagnosis as patients often have more than one, and we were more interested in other factors. However, we have added to the results the following:

“Data on diagnoses were not collected and are not available in any NZ Ministry of Health publications. However, a study based in Auckland, New Zealand found the common discharge diagnoses to be mood disorders, including bipolar disorder (manic, depressive, and mixed episodes) and depression, as well as psychotic disorders, such as schizophrenia and schizoaffective disorder.[46] Additionally, due to the high demand for acute inpatient care, in order to access a psychiatric bed in New Zealand, service users need to present with high acuity; therefore, all service users participating in this study likely had a severe mental illness.[47]“

Similarly – were the staff young or old? How long had they worked in mental health? You recruited diverse group, though don’t give a breakdown of numbers from each professional role. Had any of the staff received train in smoking cessation?

Data on ages has been added. We did not collect data on length of time working in mental health. We have now provided data on staff roles to the paper.

What was the smoking status of staff and pts? If none of the info was collected some acknowledgement should be mentioned in the limitations.

We did not ask about the smoking status of staff and patients and have noted this limitation in the discussion.

Some context about what the policy involves – eg were these hospital policies just related to the environment (which you explain nicely in table 2) or did they also include a tobacco dependence treatment pathway/protocol and if so what did this involve? eg are smokers given NRT on arrival to the ward and throughout their stay, is it prescribed regularly of do pts have to request it. Is behavioural support provided at all either by ward staff or trained tobacco dependence treatment advisors?

We have added the information we have available to the beginning of the results although this is limited to what forms of NRT are/were available. We did not collect the additional data requested by reviewer 2. 

The discussion is comprehensive. Some mention of staff education and training is pertinent when you discuss solutions.

A new paragraph has been added to the discussion on the education and training staff should receive to feel more comfortable with implementing smoke-free policies. This includes education on NRT options, interactions, and myth busting.

A couple of copy edit points

‘this MS report’s

‘Alternatively, smoking’s assumed calming effects may simply be the alleviation of

withdrawal effects.[put ref above that I deleted here]’

We have gone through and addressed these errors.

Well done on a great study.

Thank you very much.

Reviewer 3 Comments:

This is an interesting article describing tobacco smoking in smoke-free mental health units in New Zealand. The authors apply a social constructionist problem representation approach to interpretation of semi-structured interview data. The framework guiding the analysis is infrequently used in the context of smokefree mental health research, and that is a strength of this paper. 

The manuscript however, is rather wordy and lacking in some pertinent details, and I feel that this detracts somewhat from its strengths. Some points that the authors might consider:

Abstract: The stated aim of the study is to explore the smoking milieu in the mental health environment. However, the results imply the exploration of other factors in addition to the environment. It is also quite difficult to extract the conclusion from the abstract. It would help if the authors could make clear what they mean by ‘milieu’ – at present it is a little ambiguous.

We have removed the term ‘milieu’. 

Additional details on the NZ research settings has been added to the methods. 

Introduction: Some of the cited material the authors rely on was published over a decade ago, the inclusion of references to more recent published articles on the implementation and evaluation of smokefree policies would be welcome here.

We have added several more recent references to the introduction and the discussion.

It would also help to orientate international readers if a short summary of the New Zealand mental health landscape was provided. 

We have added a new section under NZ setting in the methods to address this.

Furthermore, it would be helpful to know if service users are provided with any pharmacological or behavioural support as part of the smokefree policy. 

No, they are not provided any pharmacological or behavioural support as part of the smoke free policy. We have added more about this and the available options in NZ in the discussion.

A little more detail on the research settings and locality of the findings described would also help the reader to determine contextual similarities and differences to the New Zealand context. This could also be revisited in the discussion. A large number of studies are referenced, but rather than provide a summary of the landscape the introduction would be more compelling if the evidence were used to compile a path forward and a rationale for the present study.

We are happy with the reworked introduction now. A section on NZ setting has been added to the methods and a statement on the generalisability of our findings to the rest of NZ and internationally has been added to the discussion. 

Methods: The authors provide a rather brief overview of the methods they adopted, and leaves the reader with a number of questions. For example, a purposive approach to sampling of the staff group was used, was the same applied to service users? 

For staff we used a purposive approach and for service users we had to have sign off by the lead clinician (as per our ethics requirements), so we were guided by the lead clinicians as to who was well enough for an interview. We then approached these case by case until we had interviewed around 10 from each staff and service user groups on each ward. This is noted in the methods. 

It appears that analysis comprised an iterative thematic approach. What is unclear is the actual process that was undertaken to derive the themes and subthemes. Was analysis undertaken by site/case? How were discrepancies in coding or theme generation resolved? Was a process of double coding employed? The authors should provide a full account of methods. A copy of the interview schedule as supplementary material would also be appreciated.

In the methods section, we have made clearer the process taken to derive the themes and subthemes. We have mentioned how we undertook a triple coding process and analysed data by site. We have also noted how we dealt with discrepancies in coding. We do not think it is useful to provide a copy of the interview schedule which covered multiple topics as the focus was not on smoking. 

Results: A table summarising participant characteristic and the inclusion of staff group or service user ward type (locked/unlocked) would help to locate the quotations in terms of perspective. It would also help if the frequencies and/or percentages were reported when discussing majority/minority etc. of participants.

Data on participant characteristics, including gender and ethnicity, has been refined in the new table in the manuscript. We have added additional information on the age of both staff and service users, and the occupations held by staff. We do not agree that it is necessary or useful or a good idea in qualitative research to report percentages to quantify the number of participants supporting a particular position.

Discussion: The discussion provides details of the New Zealand experience of implementing smokefree policies in acute mental health settings. Whilst the authors experience is in part very similar to that reported by other researchers in other developed countries, there are some local differences and I encourage the authors to expand on the differences; such as the predominance of locked settings and the issue of second-hand smoke in outdoor court yards. 

We cannot be sure if New Zealand has a predominance of locked settings compared with other countries as other studies have not systematically reported if their research is from locked or unlocked wards. We feel the section ‘Smoking monopolises the unit courtyard’ adequately highlights the issue of second-hand smoke in outdoor courtyards. 

It is also helpful that the authors consider the implications of their findings and make recommendations. The limitations might also make mention as to the generalisability of findings within New Zealand and more widely. Perhaps also the authors might consider the impact that those service users who were not or could not be recruited to the study may have on the findings.

We have expanded upon the implications of our research and provide guidance to the sector wishing to support smoking cessation in aonts mental health service users. We have expanded on the strengths and limitations of this study, making note of the generalisability of the findings within the New Zealand context and more widely. We prefer not to discuss the impact that services users who were not recruited may have on the study findings as it would be speculative at best. 

The inclusion of e-cigarettes in the discussion is very interesting and relevant. I would encourage the authors to expand on this, particularly in relation to its implementation in the New Zealand context. I would also encourage the authors to identity additional literature from all perspectives in relation to e-cigarettes.

We have included more discussion on e-cigarettes within New Zealand acute mental health wards. 

Conclusion: The conclusion seems to lose its way a little. Mention of the more notable findings with potential further avenues for research would I’m sure be appreciated by readers.

We have re-written the conclusion to reflect the updated revised paper.

---

## [Decision Letter · Decision Letter 1]

2 Nov 2021

There’s no smoke without fire: smoking in smoke-free acute mental health wards

PONE-D-21-12042R1

Dear Dr. Jenkin,

We’re pleased to inform you that your manuscript has been judged scientifically suitable for publication and will be formally accepted for publication once it meets all outstanding technical requirements.

Kind regards,

Michael Cummings, PhD

Academic Editor

PLOS ONE

Additional Editor Comments (optional):

Reviewers' comments:

Reviewer's Responses to Questions

**Comments to the Author**

1. If the authors have adequately addressed your comments raised in a previous round of review and you feel that this manuscript is now acceptable for publication, you may indicate that here to bypass the “Comments to the Author” section, enter your conflict of interest statement in the “Confidential to Editor” section, and submit your "Accept" recommendation.

Reviewer #2: All comments have been addressed

Reviewer #3: All comments have been addressed

2. Is the manuscript technically sound, and do the data support the conclusions?

Reviewer #2: Yes

Reviewer #3: Yes

3. Has the statistical analysis been performed appropriately and rigorously? 

Reviewer #2: N/A

Reviewer #3: N/A

4. Have the authors made all data underlying the findings in their manuscript fully available?

Reviewer #2: Yes

Reviewer #3: Yes

5. Is the manuscript presented in an intelligible fashion and written in standard English?

Reviewer #2: Yes

Reviewer #3: Yes

6. Review Comments to the Author

Reviewer #2: Dear Authors

I enjoyed reading your revised manuscript. Thanks for addressesing my commennts in the initial review

Reviewer #3: (No Response)

7. PLOS authors have the option to publish the peer review history of their article (what does this mean?). If published, this will include your full peer review and any attached files.

Reviewer #2: No

Reviewer #3: No

---

## [Editor Report · Acceptance letter]

5 Nov 2021

PONE-D-21-12042R1 

There’s no smoke without fire: smoking in smoke-free acute mental health wards 

Dear Dr. Jenkin:

I'm pleased to inform you that your manuscript has been deemed suitable for publication in PLOS ONE. Congratulations! Your manuscript is now with our production department. 

Kind regards, 

on behalf of

Dr. Michael Cummings 

Academic Editor

PLOS ONE